# Optical Biomedical Imaging Reveals Criteria for Violated Liver Regenerative Potential

**DOI:** 10.3390/cells12030479

**Published:** 2023-02-02

**Authors:** Svetlana Rodimova, Nikolai Bobrov, Artem Mozherov, Vadim Elagin, Maria Karabut, Ilya Shchechkin, Dmitry Kozlov, Dmitry Krylov, Alena Gavrina, Vladimir Zagainov, Elena Zagaynova, Daria Kuznetsova

**Affiliations:** 1Institute of Experimental Oncology and Biomedical Technologies, Privolzhsky Research Medical University, 10/1 Minin and Pozharsky Sq., 603000 Nizhny Novgorod, Russia; 2Department of Biophysics, N.I. Lobachevsky Nizhny Novgorod National Research State University, 23 Gagarina Ave., 603022 Nizhny Novgorod, Russia; 3The Volga District Medical Centre of Federal Medical and Biological Agency, 14 Ilinskaya St., 603000 Nizhny Novgorod, Russia; 4Nizhny Novgorod Regional Clinical Oncologic Dispensary, Delovaya St., 11/1, 603126 Nizhny Novgorod, Russia

**Keywords:** liver pathology, liver regeneration, multiphoton microscopy, FLIM, SHG

## Abstract

To reduce the risk of post-hepatectomy liver failure in patients with hepatic pathologies, it is necessary to develop an approach to express the intraoperative assessment of the liver’s regenerative potential. Traditional clinical methods do not enable the prediction of the function of the liver remnant. Modern label-free bioimaging, using multiphoton microscopy in combination with second harmonic generation (SHG) and fluorescence lifetime imaging microscopy (FLIM), can both expand the possibilities for diagnosing liver pathologies and for assessing the regenerative potential of the liver. Using multiphoton and SHG microscopy, we assessed the structural state of liver tissue at different stages of induced steatosis and fibrosis before and after 70% partial hepatectomy in rats. Using FLIM, we also performed a detailed analysis of the metabolic state of the hepatocytes. We were able to determine criteria that can reveal a lack of regenerative potential in violated liver, such as the presence of zones with reduced NAD(P)H autofluorescence signals. Furthermore, for a liver with pathology, there was an absence of the jump in the fluorescence lifetime contributions of the bound form of NADH and NADPH the 3rd day after hepatectomy that is characteristic of normal liver regeneration. Such results are associated with decreased intensity of oxidative phosphorylation and of biosynthetic processes in pathological liver, which is the reason for the impaired liver recovery. This modern approach offers an effective tool that can be successfully translated into the clinic for express, intraoperative assessment of the regenerative potential of the pathological liver of a patient.

## 1. Introduction

Today, in worldwide clinical practice, about 1 million new cases of primary and secondary liver cancers are recorded annually. The only treatment option for such patients is liver resection. Generally, 13–30% of patients can undergo surgical treatment, but the 5-year survival rate is only 14–61% [1,2]. Despite modern advances in surgical technique and the improvement in methods for preoperative assessment of liver function, there is currently a high risk of postoperative liver failure associated with the presence of background hepatic pathologies. 

Although there are already substantial data related to the regeneration processes of a healthy liver, in actual clinical practice, about 20% of patients undergoing liver resection, and up to 25% of donors for liver transplantation, have some degree of steatosis or fibrosis [3,4,5]. It is known that 25% is the minimum volume of remnant liver sufficient for the effective recovery of a healthy liver. By contrast, in the presence of underlying hepatic pathologies, at least 40% of the remnant liver volume could be required for effective recovery [6]. 

Non-alcoholic fatty liver disease (NAFLD) is the most frequent pathology (in 30% of the adult population in developed countries) that reduces the regenerative potential of the liver. The most common form of NAFLD is steatosis, associated with an increased intensity of biosynthesis of fatty acids and glycerol and an accumulation of triglycerides [7].

Another frequent liver disease is fibrosis, or its decompensated stage, cirrhosis. Fibrosis is accompanied by a disruption of tissue architecture, the development of portal hypertension, and of cell hypoxia. Among patients with underlying cirrhosis or severe fibrosis, the mortality rate after surgical treatment of hepatocellular carcinoma is 30% and is associated with an insufficient functional reserve and a reduced regenerative potential of the liver [8,9].

Despite continuing improvements in the methods for assessing the structure of liver tissue, histological and immunohistochemical analyses are still considered the gold standards. However, these methods are time-consuming and do not provide information about vital processes within the tissue. Indeed, the diagnosis of cirrhosis relies primarily on histopathological evidence of late-stage fibrosis (e.g., stage 4 fibrosis using the METAVIR system, or at stages 5 or 6 in the Ishak scoring system), but does not include any analysis of the pathogenesis and functional capabilities of the liver tissue [10]. 

Currently, fluorescence bioimaging methods are being actively used in biomedical research. These include multiphoton microscopy, second harmonic generation (SHG) microscopy, and fluorescence lifetime imaging microscopy (FLIM). The procedure for obtaining and processing the data resulting from such methods is not time-consuming, so this is an advantage for enabling express, intraoperative assessment of the state of the liver tissue. The main additional benefits of these methods are that they are minimally invasive, do not require additional contrast agents, and allow the intravital visualization of tissue structure at cellular resolution. Additionally, based on data on the fluorescence lifetimes of various forms of the intracellular cofactors NAD(P)H and their relative contributions, we can assess changes in the metabolic state of the hepatocytes, which is a sensitive indicator of the state of the liver tissue. Previously, we were able to show that metabolic bioimaging methods are effective for assessing the metabolic state of hepatocytes in the presence of pathologies and in the process of normal regeneration [11,12,13]. We also confirmed that by using FLIM, it is possible to define the stage of the regenerative process, based on specific changes in the intensity of oxidative phosphorylation (OXPHOS) and of biosynthetic processes. However, there is still a lack of information about the metabolic state of the hepatocytes during regeneration at different stages of underlying pathologies.

Thus, the purpose of this current work was to identify new criteria related to the reduced regenerative potential of liver with underlying pathologies by using the label-free, minimally invasive methods of multiphoton microscopy with SHG and FLIM.

## 2. Materials and Methods

### 2.1. Animal Model

A series of experiments were carried out using 30 male Wistar rats with an average weight of 300–400 g. Induction of toxic fibrosis was carried out by the intraperitoneal injection of a 33% solution of carbon tetrachloride (CCl_4_) diluted in oil twice a week for 8 weeks [14]. Alternatively, to induce steatosis, the laboratory animals were put on a high-fat diet [15], containing 60% of its calories as fat, for 12 weeks. The regeneration process was induced by 70% partial hepatectomy (70% PH) [16]. After resection, each animal was placed in a clean cage and kept under the standard conditions of an SPF vivarium. To analyze the state of the liver tissue at different stages of the hepatic pathologies before the induction of liver regeneration, we examined ex vivo liver samples taken during the resection (day 0). On the 3rd and 7th days after resection, remnant samples (whole organ) were taken from the animals for study.

### 2.2. Measurements of Liver Weight Recovery and Morphometric Analysis

To determine the effectiveness of liver recovery, the liver weight was measured before the induction of regeneration and at different stages of the recovery process. The initial liver weight was calculated according to the following formula: *weight of resected liver (g)/0.7*. The percentage of liver weight recovery was carried out according to the following formula: *weight of remnant (g)/initial liver weight (g)*—this is the absolute percentage of liver weight recovery, i.e., “absolute weight”. To assess the percentage of recovery of liver tissue with induced pathologies relative to the corresponding day of normal regeneration, we calculated the relative percentage of liver weight recovery, i.e., “relative weight”. For this purpose, the values of “absolute weight” for normal liver regeneration were taken as 100% (absolute weight norm) and divided into “absolute weight” for the liver with pathology (absolute weight path) at the corresponding time point of regeneration, i.e., the 3rd or 7th day after PH: *absolute weight_path_(g)/absolute weight_norm_(g) × 100%*.

Morphometric analysis was performed with the following indicators being evaluated: the number of tetraploid hepatocytes (cells with brightly colored, enlarged nuclei) and the numbers of binucleate cells and of mitotic cells. The average values of all these parameters were calculated in proportion to 100 normal cells [17]. The morphometric analysis was carried out using histological sections with hematoxylin and eosin and van Gieson staining. For each sample, 10 micrographs were obtained (×400) using a Leica DM 2500 microscope (Munich, Germany).

### 2.3. Biochemical Blood Tests

Blood samples (1.5 mL) were obtained from the tail vein of each rat. The blood was then centrifuged at 3000 rpm for 15 min to obtain the serum. The following parameters were evaluated: the levels of aspartate aminotransferase (AST), alanine aminotransferase (ALT), alkaline phosphatase (AP), total protein, creatinine, urea, high-density lipoproteins (HDL), low-density lipoproteins (LDL), and triglycerides (TG) [18]. The analysis was carried out using an automated biochemical analyzer (Mindray BS-120, Shenzhen, China) and standard reagents (Vital, Russia) in accordance with the manufacturer’s protocol.

### 2.4. Multiphoton Microscopy

Investigation of all the fresh liver samples was performed using an LSM 880 (Carl Zeiss, Germany) equipped with a Ti:Sapphire femtosecond laser (repetition rate: 80 MHz, pulse duration less than 100 fs) and a time-correlated single photon counting (TCSPC) system (Simple-Tau 152, Becker & Hickl GmbH, Berlin, Germany). The average laser power used was about 10 mW. A C Plan-Apochromat 40×/1.3 oil immersion objective was used to collect the fluorescence signal. From 10 fields of view for each sample, both the NAD(P)H fluorescence intensity images and the FLIM data were acquired. NAD(P)H: λex = 750 nm, λem = 450–490 nm. The SHG signal was generated at 800 nm and recorded in the 371–421 nm range. 

We performed a quantitative assessment of the NAD(P)H autofluorescence intensity in the cytoplasm of the hepatocytes by manual selection of ~40 × 40 pixel zones as regions of interest (ROIs) using ImageJ software (National Institutes of Health, Bethesda, ML, USA).

The FLIM analysis was performed using SPCImage software (Becker & Hickl GmbH, Berlin, Germany) with a tri-exponential decay model. To maintain a minimum of 10,000 counts per pixel, the binning parameter was set at 3. The goodness-of-fit model was assessed by the χ2 value (it should be at 1). The following parameters were analyzed in 20–30 regions of cell cytoplasm for each field of view: tm (ps), the amplitude weighted mean fluorescence lifetimes; t1 (ps) were fixed at 400; t2 (ps), fluorescence lifetimes of the bound form of NADH; t3 (ps), fluorescence lifetimes of NADPH; the relative contributions of the free, a1 (%), and the bound, a2 (%), forms of NADH; and the relative contribution of NADPH, a3 (%).

### 2.5. Histological Analysis

For the histological studies, the liver was fixed in a 10% solution of buffered formalin, passed through isopropyl alcohol and embedded in paraffin. Deparaffinized 7 μm sections were stained with hematoxylin/eosin and picrofuxin by van Gieson according to the standard protocol [19]. For each sample, 10 micrographs were obtained (×400) using a Leica DM 2500 microscope (Munich, Germany).

### 2.6. Statistics

To determine the statistical significance of changes in the degree of tissue regenerative capacity of the liver with induced pathology, the nonparametric Mann–Whitney U test was used. Comparative analyses were performed between recovered liver weight with induced pathology on the 3rd and 7th days after 70% PH with the corresponding time point of normal liver regeneration; *p*-value ≤ 0.05. 

For each time point of the experiment, we obtained 8–10 NAD(P)H autofluorescence and FLIM images. For each image, we performed a quantitative assessment of the NAD(P)H autofluorescence intensity in the cytoplasm of 30 hepatocytes (in zones with a high NAD(P)H autofluorescence intensity) and for 30 hepatocytes (in zones with a low NAD(P)H autofluorescence intensity), excluding the nucleus. For each image, we determined the FLIM parameters in the cytoplasm of 30 hepatocytes (in zones with a high NAD(P)H autofluorescence intensity). R-language was used for the statistical calculations. Statistical differences between different groups were analyzed using the pairwise multiple comparison procedure. Differences in mean values between groups were considered significant when they were larger than might be expected by chance. After the tests for normality (Shapiro–Wilkes) and equal variance (F-test) showed a normal distribution of data, the pairwise t-test method was used, for pairwise comparison, using the Bonferroni correction; *p*-value ≤ 0.05.

## 3. Results

The design of this study is depicted in Figure 1, where the experimental stages are presented, for normal liver regeneration, liver regeneration with induced steatosis, and liver regeneration with induced fibrosis.

### 3.1. Measurements of Liver Weight Recovery

We analyzed the restoration of liver weight at different stages during the induced regeneration process of liver with steatosis and fibrosis using standard methods [20,21,22]. To assess the effectiveness of liver recovery relative to the initial volume at different stages of the pathology, we determined the weight of the liver before 70% PH and calculated the percentage of the restored liver weight on the 3rd and 7th days after the 70% PH, i.e., “absolute weight”. In addition, we analyzed changes in liver weight at different stages of the pathologies during regeneration, relative to physiological (normal) regeneration, i.e., “relative weight”. The results are presented in Table 1.

For steatosis at the 3rd week, we observed a dramatic and significant increase in the absolute and relative weight of the recovered liver on the 3rd and 7th days, followed by a decrease in these parameters up to 12th week. At the 12th week of steatosis, the values of the absolute and relative weight were almost equal to the corresponding values for normal regeneration. Such high values are probably associated not only with the restoration of the mass of the liver parenchyma but also with the accumulation of lipid droplets and the progression of fibrosis.

For induced fibrosis, we could observe a sharp drop in both the absolute and the relative liver weights on the 3rd and 7th day after PH by the 2nd week of fibrosis induction. However, the values of the liver weight on the 3rd day after PH increased gradually from the 4th week, and by the 8th week even exceeded the corresponding values for normal regeneration. However, despite the high values of liver weight on the 3rd day after PH, on the 7th day (termination of the regeneration process) for all stages of the pathology, except for the 7th day at the 4th week, the percentage of liver recovery was already falling.

It is worth considering that our histological analysis and analysis of the structural and functional state of the liver tissue using multiphoton microscopy showed obvious pathological changes.

### 3.2. Morphometric Analysis

We observed that during the development of hepatic steatosis, there was a significant decrease in the hepatocyte proliferative activity on the 3rd day of regeneration in comparison with the model of normal regeneration (Figure 2). In the case of toxic cirrhosis, there was a sharp decrease in the hepatocyte proliferative activity in the early stages of the pathology (2nd week), followed by an increase in the number of dividing hepatocytes in the later stages (Figure 2).

### 3.3. Biochemical Blood Tests 

The results of a biochemical blood test are presented in the Appendix A. For both pathologies, we detected tissue damage and a decrease in the synthetic liver function before and after PH.

### 3.4. Analysis of the Structure of Liver Tissue

Histological analysis of the liver tissue with induced steatosis showed a pronounced lipid infiltration of the hepatocytes by the 6th week of the pathology. Starting in the 6th week, we identified portal fibrosis without septa (METAVIR fibrosis score of F1). By the 12th week, an advanced stage of steatosis was observed, characterized by macrovesicular steatosis (Figure 3a). However, the overall architecture of the tissue at each stage of induced steatosis was not significantly disturbed. In the NAD(P)H autofluorescence channel, we could recognize zones with a reduced signal of NAD(P)H autofluorescence that were associated with lipid infiltration (Figure 3a). Additionally, with the progression of steatosis, we observed a gradual increase in bright inclusions that represented stellate cells of the liver (of the type that store vitamin A) [23]. The rise in vitamin A inclusions was more pronounced in the case of steatosis than in fibrosis. 

In the model of regeneration induction after steatosis, the results of SHG analysis are consistent with the data of morphological analysis, where we could observe single foci of accumulation of collagen fibers (Figure 3a). In the NAD(P)H autofluorescence channel, we recognized zones with reduced NAD(P)H autofluorescence signals that were predominantly associated with areas of high lipid infiltration (Figure 3a).

For induced fibroses, we observed typical pathological changes that were most pronounced at the 6th and 8th weeks of the pathology. In the 6th week, we identified the presence of fibrosis with single septa (METAVIR fibrosis score of F2 or greater) and the accumulation of lipid droplets in the hepatocytes. In the 8th week, histological analysis showed foci of necrosis, the accumulation of lipid droplets in the hepatocytes, and severe fibrosis (METAVIR fibrosis score of F3 or greater) (Figure 4a). Furthermore, there was a disturbance of the liver tissue architecture in the late stages of the fibrosis (at the 6th and 8th weeks) (Figure 4a). Such a disturbance is primarily associated with a violation of the lobular architecture due to the formation of fibrous septa. The results of the histological analysis are consistent with the SHG data, where we observed foci of collagen accumulation (Figure 4a). In the NAD(P)H autofluorescence channel, at all stages of the pathology, we could recognize zones with a reduced signal, which, unlike for steatosis, were associated mostly with the foci of fibrosis (Figure 4a).

The regeneration process at different stages of fibrosis was characterized by pronounced microvesicular steatosis being seen on both the 3rd and 7th day after PH. We also observed a violation of tissue architecture on both the 3rd and 7th day after PH for each pathological stage. This was primarily due to the rate of hepatocyte proliferation being higher than the rate of synthesis of intercellular substance [24,25]. The results of the histological analysis were consistent with the SHG data, where we could observe a few foci of fibrosis by the 6th week and by the 8th week, when there was already a severe accumulation of collagen fibers (Figure 4a). Using multiphoton microscopy, by the 2nd week, we could already observe zones with a reduced NAD(P)H autofluorescence signal; however, unlike in hepatic steatosis, such zones mainly corresponded to fibrous tissues. Additionally, during the progression of fibrosis, we could observe a significant increase in bright inclusions in the liver tissue, which represented stellate cells of the liver of the type that store vitamin A.

We performed quantitative assessments of the NAD(P)H autofluorescence intensity for both pathologies in two identified zones—with high (I1) and low (I2) NAD(P)H auto-fluorescence intensities. We showed a statistically significant decrease in the NAD(P)H autofluorescence intensities in zones I2, associated with damaged hepatocytes, lipid infiltration, and fibrous tissue. 

Furthermore, in the case of hepatic steatosis, at the 3rd week of pathology, we observed a sharp, statically significant decrease in the intensity of the NAD(P)H autofluorescence in zones I1, in comparison with corresponding zones I1 for normal regeneration. Further, the intensity of NAD(P)H autofluorescence in such I1 zones gradually increased. By the 12th week, the values of the intensity of NAD(P)H autofluorescence on the 3rd and 7th days after PH were almost equal to the corresponding values in zones I1 for normal regeneration. The results are presented in Appendix A.

At the same time, in the case of fibrosis, the intensity values of NAD(P)H autofluorescence in zones I1 were statically significantly lower compared to the corresponding values in the I1 zones for normal regeneration at all experimental time points. The results are presented in Appendix A.

Thus, we found that the main criterion indicating steatosis and fibrosis was the presence of zones with a reduced NAD(P)H autofluorescence signal intensity, these being associated with foci of infiltration or foci of fibrosis.

### 3.5. FLIM Analysis

Using FLIM, we analyzed the metabolic state of the hepatocytes in liver tissue with induced pathologies before and after the induction of the regeneration process.

In the case of liver steatosis, the fluorescence lifetime values of the bound form of NADH (t2, ps) and of NADPH (t3, ps) did not change significantly at any of the monitoring time points. In the case of liver fibrosis, we revealed statistically significant drops in the values of the fluorescence lifetime of the bound form of NADH in the later stages (6th and 8th weeks) that are probably associated with hypoxia in the hepatocytes [26,27]. The results are presented in Appendix A.

An analysis of the relative contributions of various forms of the NAD(P)H showed that the values of the lifetime fluorescence contribution of the bound form of NADH (a2, %) sharply decreased in the 3rd week of hepatic steatosis. However, with further development of the pathology (by the 6th week) this parameter increased again, and by the 9th week, the values of a2 almost reached the control values (for normal regeneration) (Figure 3b,c). However, in the later stages of hepatic steatosis, the a2 values fell again. We also evaluated changes in the fluorescence lifetime contribution of NADPH (a3, %) at different stages of steatosis. It was shown that by the 3rd week, there was a sharp decrease in the a3 value. Then, over the 6th, 9th, and 12th weeks, there was a slight growth of a3, although not reaching the control values (for normal regeneration).

During the regenerative process at different stages of steatosis, it was shown that by the 3rd week, on the 3rd day after PH, there was a sharp increase in the fluorescence lifetime contribution of the bound form of NADH (a2, %) and the contribution of NADPH (a3, %), followed by a decrease on the 7th day (Figure 3b,c). Such a trend is similar to that seen during normal liver regeneration. However, by the 6th and 9th weeks, the trend had changed; specifically, there was no sharp jump in the values of a3 on the 3rd day after PH. As can be seen in Figure 3b,c, starting in the 6th week of steatosis, the a2 and a3 values were almost the same before and after PH. A similar trend subsequently persisted for a2 at both the 9th and 12th weeks.

In the case of fibrosis, we observed a gradual decrease in the values of the fluorescence lifetime contribution of the bound form of NADH (a2, %) by the 2nd and 4th weeks after the beginning of CCl_4_ administration (Figure 4b,c). Despite this, by the 6th and 8th weeks, we observed a gradual rise in the value of a2. Already by the 2nd week, it could be observed that there was a sharp decrease in the contribution of NADPH (a3, %), which could also be seen later, in the 4th and 6th weeks; however, by the 8th week, the values of a3 had returned to match the control levels.

As in the case of steatosis, we found a similar trend in the regeneration process with fibrosis. It was shown that during the early stages (in the 2nd and 4th weeks) the values of a2 and a3 before and after PH (both on 3rd and 7th days) were similar (Figure 4b,c). By the 8th week, this trend persisted for a2 (Figure 4b,c). However, we observed a slight increase in a3 on the 3rd day after PH, followed by a decrease on the 7th day (Figure 4b,c), which was probably due to increased lipogenesis.

Thus, using FLIM, we have identified several criteria related to the reduced regenerative potential of livers with pathologies. In the early stages of the pathology, decreased values of a2 and a3 were a typical change. For the later stages of the pathologies, an important aspect was the absence of a sharp increase in the contributions of a2 on the 3rd day after PH. However, the absence of a corresponding sharp jump in a3 on the 3rd day after PH was observed only in the case of steatosis. Such observations are consistent with the fact that effective regeneration requires a significant increase in the intensity of OXPHOS (high values of a2) to drive ATP synthesis and also requires an increase in the biosynthetic processes (high values of a3) that meet the energy needs of hepatocytes.

## 4. Discussion

Previously, we have shown that in a healthy liver, OXPHOS is the most active metabolic pathway in the hepatocytes. Our results showed that in the process of physiological (normal) regeneration, on the 3rd day after 70% PH, there is a sharp jump in the contribution of the bound form of NADH and NADPH that indicates an even greater increase in the intensity of OXPHOS and synthetic processes due to the high energy demands of the proliferating hepatocytes [12]. However, more data are required on the metabolic state of the hepatocytes at different stages of hepatic pathologies and at different stages of the regeneration process in the presence of pathologies. In this work, we carried out a comprehensive analysis of the structural and functional state of liver tissue at different stages of steatosis and fibrosis and with induced regeneration, using the modern methods of multiphoton microscopy with SHG and FLIM. 

It is widely known that hepatic fibrosis and cirrhosis appear to be significant risk factors for post-operative complications, disrupting hepatic regeneration [28,29]. The regenerative capacity of steatotic liver is less well established. Studies on liver regeneration with steatosis that have been performed in animal models have suggested that steatosis does also impair liver regeneration [30,31]. In humans, such studies are scarce. Most of the few existing reports have focused on regeneration in living donors for liver transplantation, but the results are contradictory [32]. Indeed, such studies were mainly based on clinical methods and the results of biochemical blood tests. Such methods do not allow for the evaluation of any changes in liver function, which can be impaired even if the liver mass is completely restored. 

In our work, we confirmed that morphological methods are not sufficiently specific and informative; therefore, we tried to determine specific features of the metabolic state of hepatocytes during steatosis and fibrosis undergoing induced regeneration. The metabolic state of the hepatocytes is a more sensitive marker of the presence of pathological processes at the level of individual cells, while also being associated with liver function even at the level of the whole organ [33]. Thus, the newly identified imaging criteria for the assessment of the metabolic state of hepatocytes are promising for the express assessment of the regenerative potential of liver tissue in clinical settings. Using our approach based on fluorescence bioimaging, these methods make it possible to supplement or even substitute such laborious, time-consuming, and less specific methods as histological analysis and biochemical blood tests.

Using multiphoton microscopy, we have shown that the existence of pathology can be identified by the presence of zones with reduced NAD(P)H autofluorescence signals, these being associated with zones of lipid infiltration and with the foci of fibrosis. However, such structural analysis does not provide detailed information about the metabolic processes occurring in the cells. To obtain more detailed data on the metabolic state of the hepatocytes, we performed FLIM analysis. FLIM allows us to distinguish the short- and long-lifetime component forms of NAD(P)H that reflect whether this cofactor is in its free or protein-bound state, respectively. The free form of NADH drives ATP production in the cytosol by glycolysis, while the bound form of NADH is localized in the mitochondria and is involved in OXPHOS [34]. The reduced phosphorylated form, NADPH, is involved in the biosynthesis of fatty acids and steroids, in the pentose phosphate pathway, and in antioxidation defense reactions (glutathione metabolism) [35,36]. In our work, we revealed that there was a decrease in the contributions of the bound form of NADH and NADPH even during the early stages of steatosis. Moreover, we found that with the development of steatosis, there was no increase in OXPHOS in the proliferating hepatocytes, even though it is necessary for effective, healthy regeneration. Such results are probably due to the mitochondrial dysfunction characteristic of steatosis, accompanied by a decrease in mitochondrial ATP synthesis [37]. The mitochondrial dysfunction is thought to be due to structural changes seen in the mitochondrial matrix in electron microscopic studies, the total ATP synthesis being decreased in the fatty hepatocytes because of decreased mitochondrial ATP synthase activity. Furthermore, the ability to recover the depleted hepatic ATP storage is severely impaired in steatosis [37,38].

Despite the fact that NADPH is involved in the process of lipogenesis, in our work, we did not observe an increase in NADPH contribution in the hepatocytes, either in the case of hepatic steatosis or during the liver regeneration process, even in the later stages of steatosis. These results indicate damage to the hepatocytes, leading to a general decrease in their synthetic activity. Since it is known that mitochondrial dysfunction impairs fat homeostasis in the liver and leads to an overproduction of reactive oxygen species (ROS), this could explain why increased oxidative stress and lipid peroxidation have been identified in the literature as the most prominent pathogenic features of injury in steatosis [37]. Eventually, oxidative stress may cause damage at a cellular level, such as membrane lipid peroxidation, cell degeneration, and necrosis [39]. Therefore, the drop in NADPH is probably associated with glutathione depletion in the hepatocytes.

For induced fibrosis, FLIM analysis showed a sharp decrease in the fluorescence lifetime contributions of the bound forms of NADH and NADPH in the early stages of pathology. The sharp drop in the bound form of NADH during the early stages is probably associated with mitochondrial dysfunction, which is common for fibrosis [40,41]. It is known that when CCl_4_ enters an animal’s body, it is activated by the liver cytochrome P450 to generate the free radicals CCl_3_·and CCl_3_O_2_· [40], causing the peroxidation of fatty acids in the mitochondrial membrane and thus impairing the integrity and stability of the mitochondrial structure, leading to mitochondrial dysfunction [42]. Additionally, a drop in the bound form of NADH indicates a decrease in the synthetic activity of the hepatocytes, due to toxic damage in the early stages of fibrosis. However, in our experiments, at later stages, the contributions of these bound forms of NADH and NADPH became even higher than the control values (normal regeneration). Such a result indicates the activation of compensatory mechanisms, in particular, the compensatory formation of giant mitochondria in response to an increase in the number of dysfunctional mitochondria [42]. Moreover, such an increase in the contributions of NADH and NADPH may be associated with the chronic formation of regenerative nodules surrounded by fibrotic septa [43] and activation of an antioxidant response, in particular, the glutathione cycle, in which NADPH is involved [28,36]. Our group has also previously demonstrated increases in the bound form of NADH in the presence of fibrosis, both in rat models and in ex vivo patient samples [11,12]. Our results are consistent with those of other authors. In particular, Nishikawa et al. showed that to maintain energy homeostasis, ATP production switches from predominantly involving oxidative phosphorylation to being predominantly from glycolysis in the early stages of pathology. However, the maintenance of energy production by this compensatory mechanism fails in hepatocytes in the later stages of chronic liver injury and is associated with hepatic failure [44]. Any increase in hepatocyte proliferation involves constant high energy consumption, which is probably the reason for the absence of a jump in the contribution of the bound form of NADH in the regeneration process during the late stages of fibrosis. This result once again shows that the level of hepatocyte proliferation does not correlate with the functional capacity of the hepatocytes.

## 5. Conclusions

Using an approach based on label-free methods of multiphoton microscopy in combination with SHG and FLIM, we have identified criteria for indicating reduced liver regenerative potential in the presence of steatosis and fibrosis. These criteria are as follows: 1. the presence of zones with reduced NAD(P)H autofluorescence signals, corresponding to zones of lipid infiltration and to foci of fibrosis; 2. low values of the contributions of the fluorescence lifetimes of the bound form of NADH and NADPH, due to the development of mitochondrial dysfunction and damage to the hepatocytes; 3. the absence of sharp increases in the contributions of the fluorescence lifetimes of the bound forms of NADH and NADPH in proliferating hepatocytes, associated with the depletion of energy resources in the pathological liver, a situation that significantly reduces the efficiency of liver restoration. Thus, determining the state of the above criteria will expand the possibilities for assessing the regenerative potential of a liver remnant subject to the presence of various pathologies.

## Figures and Tables

**Figure 1 cells-12-00479-f001:**
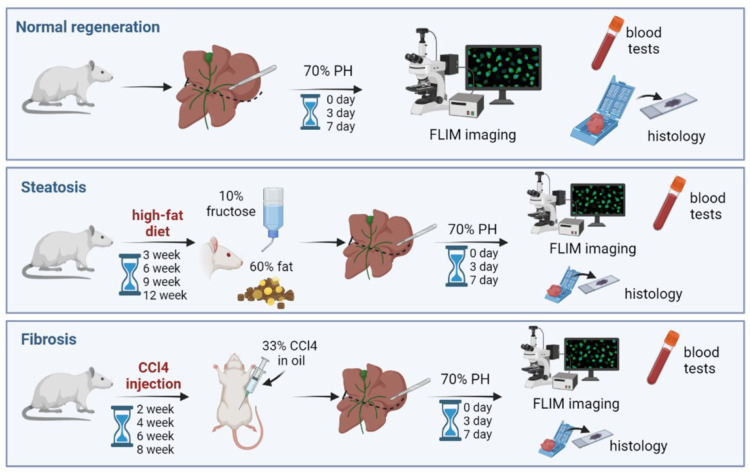
Road map of the steps of the experiment (normal liver regeneration: upper row; liver regeneration with induced steatosis: middle row; liver regeneration with induced fibrosis: bottom row).

**Figure 2 cells-12-00479-f002:**
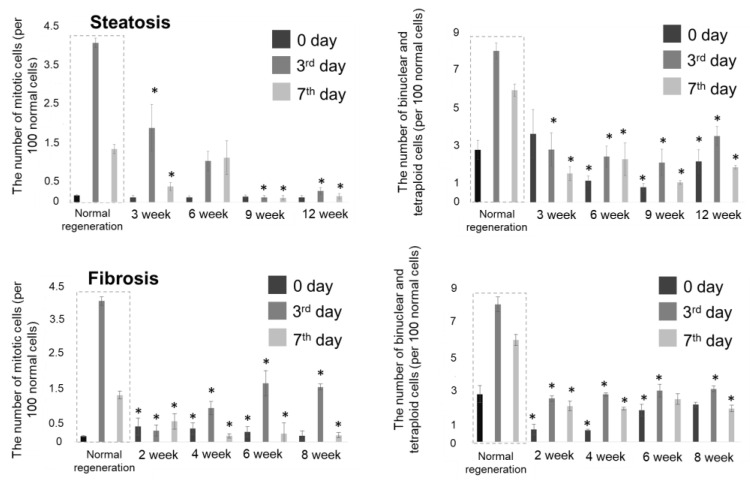
Morphometric analysis of liver tissue at different stages of normal regeneration and regeneration with steatosis and fibrosis. The charts show the number of tetraploid and binucleate cells during regeneration and the number of mitotic cells during regeneration. Values are presented as the number of dividing cells per 100 non-dividing cells. Mean ± SD. *—statistically significant difference compared to the corresponding time point for normal regeneration, *p*-value ≤ 0.05.

**Figure 3 cells-12-00479-f003:**
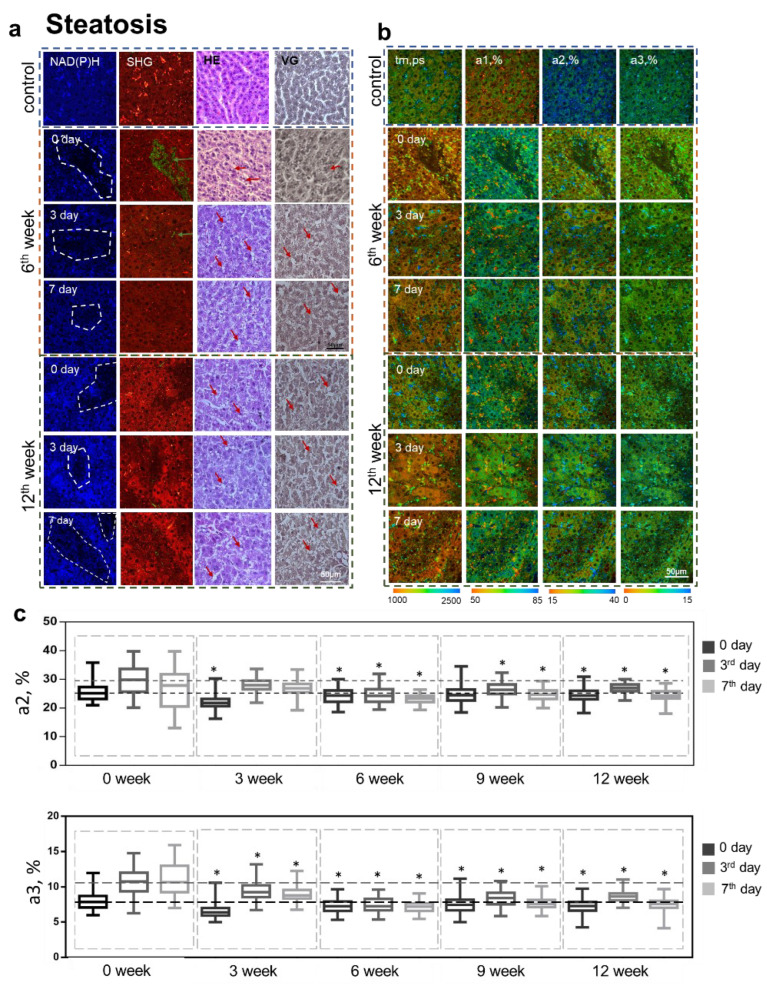
Analysis of the structural and functional state of liver tissue at different stages of steatosis during induced regeneration. (**a**) Fluorescence intensity images of NAD(P)H autofluorescence (leftmost column); NAD(P)H: excitation at 750 nm, detection range 455–500 nm. Second left column: cell autofluorescence (red): excitation at 800 nm, detection range 433–660 nm, SHG of collagen (green): excitation at 800 nm, detection range 371–421 nm; green arrows indicate collagen. Histological images: hematoxylin and eosin (HE) (second right column) and Van Gieson (VG) (rightmost column) staining; red arrows indicate lipid droplets. (**b**) Pseudo-coded FLIM images at different stages of steatosis during induced regeneration: before PH (0 day) and on the 3rd and 7th days after PH. (**c**) Boxplots reflecting the distribution of the values of the fluorescence lifetime contributions of the bound form of NADH and NADPH. The dotted line marks the medians of the parameter values for normal regeneration; scale bar 50 μm; ×400. *—statistically significant differences compared to the corresponding time point for normal regeneration, *p*-value ≤ 0.05.

**Figure 4 cells-12-00479-f004:**
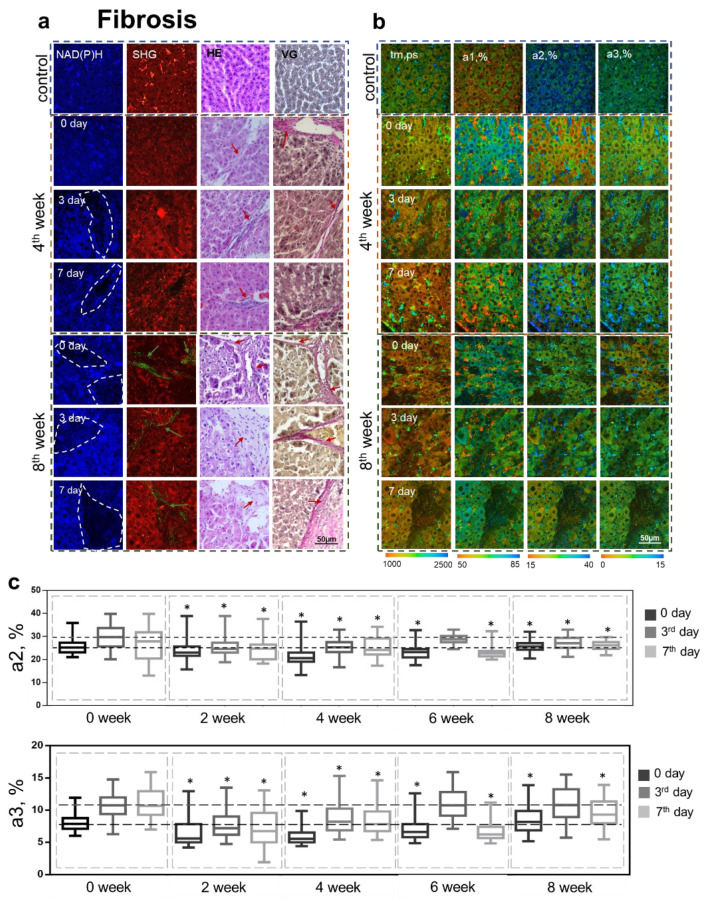
Analysis of the structural and functional state of liver tissue at different stages of fibrosis during induced regeneration. (**a**) Fluorescence intensity images of NAD(P)H autofluorescence (leftmost column); NAD(P)H: excitation at 750 nm, detection range 455–500 nm. Second left column: cell autofluorescence (red): excitation at 800 nm, detection range 433–660 nm, SHG of collagen (green): excitation at 800 nm, detection range 371–421 nm; green arrows indicate collagen. Histological images, hematoxylin and eosin (HE) (second right column) and Van Gieson (VG) (rightmost column) staining; red arrows indicate lipid droplets. (**b**) Pseudo-coded FLIM images at different stages of steatosis during induced regeneration: before PH (0 day) and on the 3rd and 7th days after PH. (**c**) Boxplots reflecting the distribution of the values of the fluorescence lifetime contributions of the bound form of NADH and NADPH. The dotted line marks the medians of the parameter values for normal regeneration; scale bar 50 μm; ×400. *—statistically significant differences compared to the corresponding time point for normal regeneration, *p*-value ≤ 0.05.

**Table 1 cells-12-00479-t001:** Analysis of the recovery of liver weight.

**Steatosis**	**Normal Regeneration**		**3 Week**		**6 Week**		**9 Week**		**12 Week**	
	**3rd Day**	**7th Day**	**3rd Day**	**7th Day**	**3rd Day**	**7th Day**	**3rd Day**	**7th Day**	**3rd Day**	**7th Day**
absolute weight (%)	80.9	92.9	105 *	136.8 *	77.1	108.6 *	80.7	84.8 *	77.4	95.4
relative weight (%)	-	-	119.3	142.7	87.6	113.2	91.7	88.4	88.7	99.54
**Fibrosis**	**Normal Regeneration**		**2 Week**		**4 Week**		**6 Week**		**8 Week**	
	**3rd Day**	**7th Day**	**3rd Day**	**7th Day**	**3rd Day**	**7th Day**	**3rd Day**	**7th Day**	**3rd Day**	**7th Day**
absolute weight (%)	80.9	92.9	71.9	60.1 *	73.1	93.3	83.7	73.6 *	111.1*	82.1 *
relative weight (%)	-	-	81.7	62.7	83	97.3	95.1	76.8	126.2	85.6

*—statistically significant difference compared to the corresponding time point for normal regeneration, *p*-value ≤ 0.05.

## Data Availability

All relevant data are found within the paper and its additional file.

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
