# Peer review of "Optical Biomedical Imaging Reveals Criteria for Violated Liver Regenerative Potential"

_cells, 2023, doi:10.3390/cells12030479_

Round 1

Reviewer 1 Report (Previous Reviewer 1)

All the comments are answered and typos are corrected. I think the revised manuscript is ready for publication.

Author Response

General comment: All the comments are answered and typos are corrected. I think the revised manuscript is ready for publication.

Reply: We thank the Reviewer for a positive evaluation of our work, and are very encouraged by your recommendation for publication in ‘Cells’.

Reviewer 2 Report (New Reviewer)

 In this manuscript, Rodimova et. Al. studied liver regeneration with fluorescence imaging methods. They compared liver regeneration at different stages of steatosis and fibrosis. They try to identify criteria for reduced liver regeneration potential in the presence of steatosis and fibrosis. The authors presented FLIM measurements and tried to extract NADH, NADPH information through a three-exponential fit. They observed different fitted values before partial hepatectomy and at different stages during recovery. However, the description of the experimental results and procedure is not clear, making it hard to judge its scientific soundness.

1. There are many presentation problems.

(1) In Table I, there is no statistical uncertainty in the data.

(2) In the main figures (3 and 4), there are inaccurate presentations.

                (a) In figures 3a and 4a, the color is different from what the legend indicates.

                (b) In 3b and 4b, what is the unit? What is control: no treatment at day 0, 3, or 7?

                (c) What is the ROI in 3a, 4a and how is it used?

                (d) It is unclear how the authors draw the conclusion from the images. For increased autofluorescence in 3a and 4a, where is the measurement? How do 3b and 4b show the changes in lifetime components in 3c and 4c? In 3C and 4C, how are the statistics done? What should be really compared to draw conclusions?

2. There are numerous grammar problems, and incomplete or incoherent sentences, which make it hard to follow. For example,

Page 1

Lines 25-27, confusing sentence

Line 38: “is remain”

Line 39: add “is” before “only”.

Page 2:

Line 38

Page 5, Lines 5-7

3. I would suggest just stating the experimental data in the main result section, leaving the plausible explanation (such as dysfunctional mitochondria function) in the Discussion.

Author Response

General comment: In this manuscript, Rodimova et. al. studied liver regeneration with fluorescence imaging methods. They compared liver regeneration at different stages of steatosis and fibrosis. They try to identify criteria for reduced liver regeneration potential in the presence of steatosis and fibrosis. The authors presented FLIM measurements and tried to extract NADH, NADPH information through a three-exponential fit. They observed different fitted values before partial hepatectomy and at different stages during recovery. However, the description of the experimental results and procedure is not clear, making it hard to judge its scientific soundness.

  1. There are many presentation problems.

Part 1. In Table I, there is no statistical uncertainty in the data.

Reply: We thank the Reviewer for the helpful remark. We carried out a statistical analysis using the Mann–Whitney U test. Appropriate additions have been made to the main text in section 2.6. ‘Statistics’.

 “To determine the statistical significance of changes in the degree of tissue regenerative capacity of the liver with induced pathology, the nonparametric Mann–Whitney U test was used. Comparative analyses were performed between recovered liver weight with induced pathology on the 3rd and 7th days after 70% PH with the corresponding time point of  normal liver regeneration; p-value ≤ 0.05.”

Corresponding changes have also been made in Table 1, and the statistical significance has been determined.

Part 2. In the main figures (3 and 4), there are inaccurate presentations.

                (a) In figures 3a and 4a, the color is different from what the legend indicates.

Reply: We thank the Reviewer for this comment. In Figures 3a and 4a, the leftmost column of images represent NAD(P)H autofluorescence images (NAD(P)H excitation at 750 nm, detection range 455–500 nm), the images in the second left column were obtained with excitation at 800 nm, cell autofluorescence being detected in the range of 433–660 nm (marked in red). In the same images (second left column), collagen fibers are marked in green.

The legends of figures 3 and 4 have been changed.

Figure 3. Analysis of the structural and functional state of liver tissue at different stages of steatosis during induced regeneration. (a)  Fluorescence intensity images of NAD(P)H autofluorescence (leftmost column); NAD(P)H: excitation at 750 nm, detection range 455–500 nm. Second left column: cell autofluorescence (red): excitation at 800 nm, detection range 433–660 nm, SHG of collagen (green): excitation at 800 nm, detection range 371–421 nm; green arrows indicate collagen. Histological images, hematoxylin and eosin (HE) (second right column) and Van Gieson (VG) (rightmost column) staining; red arrows indicate lipid droplets; (b) Pseudo-coded FLIM images at different stages of steatosis during induced regeneration: before PH (0 day) and on the 3rd and 7th days after PH; (c) Boxplots reflecting the distribution of the values of the fluorescence lifetime contributions of the bound form of NADH and NADPH. The dotted line marks the medians of the parameter values for normal regeneration; scale bar 50 μm; x400. * - statistically significant differences compared to the corresponding time point for normal regeneration, p-value ≤0.05.

Figure 4. Analysis of the structural and functional state of liver tissue at different stages of fibrosis during induced regeneration. (a) Fluorescence intensity images of NAD(P)H autofluorescence (leftmost column); NAD(P)H: excitation at 750 nm, detection range 455–500 nm. Second left column:  cell autofluorescence (red): excitation at 800 nm, detection range 433–660 nm, SHG of collagen (green): excitation at 800 nm, detection range 371–421 nm; green arrows indicate collagen. Histological images, hematoxylin and eosin (HE) (second right column) and Van Gieson (VG) (rightmost column) staining; red arrows indicate lipid droplets; (b) Pseudo-coded FLIM images at different stages of steatosis during induced regeneration: before PH (0 day) and on the 3rd and 7th days after PH; (c) Boxplots reflecting the distribution of the values of the fluorescence lifetime contributions of the bound form of NADH and NADPH. The dotted line marks the medians of the parameter values for normal regeneration; scale bar 50 μm; x400. * - statistically significant differences compared to the corresponding time point for normal regeneration, p-value ≤0.05.

                (b) Part 1. In 3b and 4b, what is the unit?

Reply: We thank the Reviewer for this comment.

We have provided the units for FLIM parameters in the ‘Materials and Methods’ section in clause ‘2.4. Multiphoton microscopy’.

 “tm (ps), the amplitude weighted mean fluorescence lifetimes; t1 (ps) were fixed at 400; t2 (ps), fluorescence lifetimes of the bound form of NADH; t3 (ps), fluorescence lifetimes of NADPH; the relative contributions of the free, a1 (%), and the bound, a2 (%), forms of NADH, and the relative contribution of NADPH, a3 (%).”

We also provided the units for the FLIM parameters on the top row of pictures (control). At the bottom of Figures 3b and 4b, a color scale represents the range of values of the corresponding parameter (tm, a1, a2 or a3) with the same units as the analyzed parameter.

 (b) Part 2. What is control: no treatment at day 0, 3, or 7?

Reply: We thank the Reviewer for this comment. In Figures 3a,b and 4a,b, normal, untreated liver tissue samples were shown as controls. However, in order to carry out the statistical analysis that we have shown in the Figures 3c and 4c, for liver tissue samples with pathology on the 3rd and 7th days of regeneration, tissue samples of normal (healthy) liver tissue on the 3rd and 7th days of regeneration (respectively) were used as controls.

                (c) What is the ROI in 3a, 4a and how is it used?

We thank the Reviewer for this helpful remark. We performed a quantitative assessment of the NAD(P)H autofluorescence intensity in the cytoplasm of the hepatocytes by manual selection of ~40 × 40 pixel zones as regions of interest (ROIs), using ImageJ software (National Institutes of Health, USA). Corresponding changes have been made to the main text in the ‘Materials and Methods’ section, ‘2.4. Multiphoton microscopy’.

Also, corresponding changes have been made in section 2.6. ‘Statistics’.

“For each time point of the experiment, we obtained 8–10 NAD(P)H autofluorescence and FLIM images. For each image, we performed a quantitative assessment of the NAD(P)H autofluorescence intensity in the cytoplasm of 30 hepatocytes (in zones with a high NAD(P)H autofluorescence intensity) and for 30 hepatocytes (in zones with a low NAD(P)H autofluorescence intensity), excluding the nucleus. For each image, we determined the FLIM parameters in the cytoplasm of 30 hepatocytes (in zones with a high NAD(P)H autofluorescence intensity).”.

It is important to note that the FLIM parameters were analyzed only in the cytoplasm of hepatocytes in zones with a high NAD(P)H autofluorescence intensity (zones I1). In these zones, hepatocytes retained high metabolic activity. In areas with a low NAD(P)H autofluorescence intensity (zones I2), analysis was not possible due to an insufficient level of collected photons.

In the main test in the ‘Results’ section, ‘3.4.’Analysis of the structure of liver tissue’ we have added the results of quantification assessment of the NAD(P)H autofluorescence intensity.

 “We performed quantitative assessments of the NAD(P)H autofluorescence intensity for both pathologies in two identified zones—with high (I1) and with low (I2) NAD(P)H autofluorescence intensities. We showed a statistically significant decrease in the NAD(P)H autofluorescence intensities in zones I2, associated with damaged hepatocytes, lipid infiltration, and fibrous tissue.

Furthermore, in the case of hepatic steatosis, at the 3rd week of pathology, we observed a sharp, statically significant decrease in the intensity of the NAD(P)H autofluorescence in zones I1, in comparison with corresponding zones I1 for normal regeneration. Further, the intensity of NAD(P)H autofluorescence in such I1 zones gradually increased. By the 12th week, the values of the intensity of NAD(P)H autofluorescence on the 3rd and 7th days after PH were almost equal to the corresponding values in zones I1 for normal regeneration. The results are presented in Table S2.

At the same time, in the case of fibrosis, the intensity values of NAD(P)H autofluorescence in zones I1 were statically significantly lower compared to the corresponding values in the I1 zones for normal regeneration at all experimental time points. The results are presented in Table S3.”.

The data on quantitative assessment of the NAD(P)H autofluorescence intensity in zones with high (I1) and low (I2) NAD(P)H autofluorescence intensity, are presented in the supplementary materials (Tables S2 and S3 in Supplementary Materials).

                (d) It is unclear how the authors draw the conclusion from the images. For increased autofluorescence in 3a and 4a, where is the measurement? How do 3b and 4b show the changes in lifetime components in 3c and 4c? In 3C and 4C, how are the statistics done? What should be really compared to draw conclusions?

Reply: We thank the Reviewer for the considered remarks.

We provided information on the quantitative assessment of the NAD(P)H autofluorescence intensity in two zones as a response to the previous comment.

In the case of FLIM analysis, Figures 3b and 4b are color-coded images, where each pixel encodes the value of a particular FLIM parameter. This image is a visual example of the distribution of the FLIM parameter values in liver tissue, confirming the correctness of our analysis. Figures 3c and 4c present a quantitative assessment of the FLIM parameters, using SPCImage software; ROIs were manually isolated in the cytoplasm of hepatocytes and the values of the parameters tm (ps), a1 (%), a2 (%), a3 (%) were obtained. For each animal, 8–10 images were obtained, and 30 ROIs were analyzed for each image. In order to draw a conclusion about the change in the metabolic state of the hepatocytes, it is necessary to compare the values of the FLIM parameters for the normal liver before the induction of regeneration (day 0) and on the 3rd and 7th days of regeneration (after 70% PH) with the corresponding time point for liver with pathology i.e. before the induction of regeneration (day 0) and on the 3rd and 7th days of regeneration.

Statistics: R language was used for the statistical calculations. Statistical differences between different groups were analyzed using the pairwise multiple comparison procedure. Differences in the mean values between groups were considered significant when they were larger than might be expected by chance; p-value ≤ 0.05. After the tests for normality (Shapiro-Wilkes) and equal variance (F-test) showed a normal distribution of data, the pairwise t-test method was used, for pairwise comparison, using the Bonferroni correction.

  1. 2. There are numerous grammar problems, and incomplete or incoherent sentences, which make it hard to follow. For example,

Page 1

Lines 25-27, confusing sentence

Reply: The corresponding sentence has been corrected.

“Furthermore, for the liver with pathology, there was an absence of the jump in the fluorescence lifetime contributions of the bound form of NADH and NADPH at 3rd day after hepactectomy that is characteristic of normal liver regeneration.”

Line 38: “is remain”

Line 39: add “is” before “only”.

Reply: Appropriate corrections have been made.

Page 2:

Line 38

Page 5, Lines 5-7

Reply: We have checked and corrected the spelling, grammar, and typographical errors in the whole text of the manuscript.

  1. I would suggest just stating the experimental data in the main result section, leaving the plausible explanation (such as dysfunctional mitochondria function) in the Discussion.

Reply: We thank the Reviewer for the helpful remark. Corresponding changes have been made to the ‘Results’ section.

Reviewer 3 Report (New Reviewer)

The main purpose of this manuscript is an assessment of the liver’s regenerative potential, not only by conventional methods, e.g. histology or blood analysis, but also by modern multiphoton microscopy, e.g. second harmonic generation (SHG) or fluorescence lifetime imaging microscopy (FLIM). Since the latter methods are label-free and minimal invasive, they have the potential to complement or in some cases even replace conventional methods. The paper describes a study with rats with induced steatosis or fibrosis. While the experimental protocol is well described (e.g. in Fig. 1), some further information is missing and should be added in a revised version of this manuscript.

Details:

- The authors should think about a more general title, since this paper does not exclusively describe FLIM experiments

- p.1, line 37-38: check grammar!

- p.2, line 48 (and later in the text): What does 70% PH mean?

- p.3, line 36: Do the authors have any idea, which molecular species is responsible for the SHG signal?

p. 3, line 43: Give a reference for the assignment of a3 to NADHP. Is it total or bound NADPH, since later (p. 10, line 28) it means bound NADPH?

p.5, Table 1: Which is the reference for “relative weight”?

p.6, Fig. 2: How did the authors determine the percentage of mitotic or binuclear cells? Visually or did they use any computer program? This Figure would need some additional explanation.

p.7, Fig. 3b: The authors should describe (Tau)m (probably an average fluorescence lifetime). How is it determined? Why don’t they specify the individual fluorescence lifetimes? In Fig. 3c it would be good to have a1 in addition to a2 and a3.

p.8-9, Fig. 4: Same comments as for Fig.3.

Author Response

General comment: The main purpose of this manuscript is an assessment of the liver’s regenerative potential, not only by conventional methods, e.g. histology or blood analysis, but also by modern multiphoton microscopy, e.g. second harmonic generation (SHG) or fluorescence lifetime imaging microscopy (FLIM). Since the latter methods are label-free and minimal invasive, they have the potential to complement or in some cases even replace conventional methods. The paper describes a study with rats with induced steatosis or fibrosis. While the experimental protocol is well described (e.g. in Fig. 1), some further information is missing and should be added in a revised version of this manuscript.

Comment 1: Details:

- The authors should think about a more general title, since this paper does not exclusively describe FLIM experiments

Reply: We thank the Reviewer for the helpful suggestion. The title of the article has been changed.

“Optical biomedical imaging reveals criteria for violated liver regenerative potential”.

- p.1, line 37-38: check grammar!

Reply: We thank the Reviewer for the helpful remark. Corresponding corrections have been made to the main text.

- p.2, line 48 (and later in the text): What does 70% PH mean?

Reply: We thank Reviewer for an important question. A 70% hepatectomy is a surgical procedure that involves the removal of 70% of the liver. During this procedure, the left lateral and medial lobes of the liver are removed. These liver lobes have a specific weight and correspond to 70% of the total weight of the liver. This procedure is a gold standard for the induction of liver regeneration [Higgins, G. M. 1931], is well tolerated by animals and allows standardized results to be obtained.

- p.3, line 36: Do the authors have any idea, which molecular species is responsible for the SHG signal?

Reply: We thank the Reviewer for this comment. Collagen I is mostly heterotrimeric, non-centrosymmetric and the most abundant form of collagen in the tissues. This technique although widely employed, is pathologist dependent and cannot be automated. SHG method have also been widely used to image collagen fibrils. The non-centrosymmetric structure of some collagen fibers give rise to SHG signals and can be used for imaging. The caveat is that SHG cannot be used for either the non-fibrous or for symmetric fibrous collagen samples. Amongst the fibril forming collagens, collagens I and II result in the strongest SHG signals while, collagen III, although fibrous, results in very weak SHG signals [Ranjit, S., et al,., 2015]. It is important to note that, the most-characterized aspect of liver fibrosis is an increases in expression and deposition of type I collagen [Tsukada, S., et al., 2006; Fuchs, B. C., et al., 2013].

- p. 3, line 43: Give a reference for the assignment of a3 to NADHP. Is it total or bound NADPH, since later (p. 10, line 28) it means bound NADPH?

Reply: We thank the Reviewer for an important question. Blacker et al. were the first to identify the third component in the total NAD(P)H fluorescence lifetime. This is the phosphorylated form of NADPH [Blacker T. S., et al. 2014]. The NADPH identified by the authors was bound to different enzymes, in particular to various dehydrogenases. Thus, this is a bound form of NADPH.

This reference has been added to the article.

-p.5, Table 1: Which is the reference for “relative weight”?

Reply: The method for estimating the relative weight was developed by the authors of the current work, based on a well-known formula [Marsman, H. A., et al., 2013]. According to the formula, we were able to obtain the percentage of liver weight recovery for a normal liver (without pathology) and at different stages of pathology (steatosis and fibrosis). Further, in order to compare how the parameter “percentage of liver weight recovery” changes in the case of pathology compared with normal liver regeneration, we calculated the “relative percentage of liver weight recovery”. For this, the values of the percentages of weight recovery of normal liver on the 3rd and 7th days after 70% PH were taken as 100% for each recovery period. Next, we calculated the percentage of the restored liver weight at different stages of the pathology on 3rd and 7th days after 70% PH. Finally, we calculated the ratio of the absolute percentage of weight recovery of normal liver to the absolute percentage of weight recovery of liver with pathology.

The final formula is presented in the ‘Materials and Methods’ section in ‘2.2 Morphological and morphometric analysis’.

-p.6, Fig. 2: How did the authors determine the percentage of mitotic or binuclear cells? Visually or did they use any computer program? This Figure would need some additional explanation.

Reply: We thank the Reviewer for an important question. To assess the number of mitotic and binucleate cells, a standard morphological analysis was performed [R. Popescu, et al., 2012], with the following indicators being evaluated: the number of tetraploid hepatocytes (cells with a brightly colored, enlarged nucleus), the number of binucleate cells and of mitotic cells. The average values of all these parameters were calculated proportionally per 100 non-dividing cells. The study was carried out on histological sections (stained with hematoxylin and eosin). For each sample, 10 micrographs were obtained (x400) using a Leica DM 2500 microscope. The evaluation was performed visually by an experienced morphologist.

-p.7, Fig. 3b: The authors should describe (Tau)m (probably an average fluorescence lifetime). How is it determined? Why don’t they specify the individual fluorescence lifetimes? In Fig. 3c it would be good to have a1 in addition to a2 and a3.

Reply: We understand the Referee’s concern.

The parameters a1, a2 and a3 are related to each other according to the formula: a1 + a2 + a3 = 100%. Parameter a1 is the relative  contribution of the fluorescence lifetime of the free form of NADH, which is involved in glycolysis [Schaefer P. M., et al. 2019]. Unlike stem cells [Xu, X., et al., 2013] or tumor cells [Ciavardelli, D., et al., 2014], where glycolysis (a1) is the main source of ATP, in hepatocytes the activity of glycolysis has only a small impact on the overall energy metabolism, while the main metabolic energy pathway in hepatocytes is oxidative phosphorylation [Nishikawa, T., et al., 2014], in which the bound form of NADH is involved (a2). In this regard, we believe that the a2 parameter is more informative. The a3 parameter is also indicative, since hepatocytes are highly synthetically active cells and the contribution of NADPH cannot be neglected. We have previously shown that the most accurate model for the analysis of hepatocyte FLIM data is the tri-exponential decomposition model of the NAD(P)H fluorescence decay curve [Rodimova S.A., et al., 2020], which allows FLIM analysis to be carried out taking into account the contribution of the third component—NADPH. Thus, we analyzed the FLIM data based on the approach we developed earlier. 

We also determined values for the fluorescence lifetimes of different forms of NAD(P)H. The ‘Materials and Method’s section has been corrected.

“The following parameters were analyzed in 20–30 regions of cell cytoplasm for each field of view: tm (ps), the amplitude weighted mean fluorescence lifetimes; t1 (ps) were fixed at 400; t2 (ps), fluorescence lifetimes of the bound form of NADH; t3 (ps), fluorescence life-times of NADPH; the relative contributions of the free, a1 (%), and the bound, a2 (%), forms of NADH, and the relative contribution of NADPH, a3 (%).”

In the ‘Results’ section, we provided a description of our observations.

“In the case of liver steatosis the fluorescence lifetime values of the bound form of NADH (t2, ps) and of NADPH (t3, ps) did not change significantly at any of the monitoring time points. In the case of liver fibrosis, we revealed statistically significant drops in the values of the fluorescence lifetime of the bound form of NADH in the later stages (6th and 8th weeks) that are probably associated with hypoxia in the hepatocytes [26,27]. The results are presented in Table S1.”

The values for t2 (ps) and t3 (ps) are presented in Table S1 in the ‘Supplementary Material’.

-p.8-9, Fig. 4: Same comments as for Fig.3.

Reply: Relevant explanations have been given in the answer to the previous comment.

General comment: The authors have performed several imaging techniques such as SHG, fluorescence intensity imaging, and fluorescence lifetime imaging (FLIM) to assess the liver's regenerative potential. I feel that the experiments are carefully performed, and the data showed how steatosis and fibrosis treatment affect the regenerative performance of the liver. 

Reply: We thank the Reviewer for the very reasonable suggestions and comments, and have tried to improve our work in accordance with all the recommendations.

Reviewer 4 Report (New Reviewer)

The authors have performed several imaging techniques such as SHG, fluorescence intensity imaging, and fluorescence lifetime imaging (FLIM) to assess the liver's regenerative potential. I feel that the experiments are carefully performed, and the data showed how steatosis and fibrosis treatment affect the regenerative performance of the liver.  

However, it seems that FLIM did not provide sufficient (reliable) information for establishing new criteria for liver regenerative potential. Thus, I think that the title of this manuscript exaggerates the content. I think that the author should carefully choose the sentence of the manuscript title and reconsider the overall story as well as conclusion.

Author Response

General comment: However, it seems that FLIM did not provide sufficient (reliable) information for establishing new criteria for liver regenerative potential. Thus, I think that the title of this manuscript exaggerates the content. I think that the author should carefully choose the sentence of the manuscript title and reconsider the overall story as well as conclusion.

Reply: We thank the Reviewer for the helpful remark. The title of the work has been changed.

“Optical biomedical imaging reveals criteria for violated liver regenerative potential”.

Round 2

Reviewer 4 Report (New Reviewer)

This version is sufficient to be published in Cells.

This manuscript is a resubmission of an earlier submission. The following is a list of the peer review reports and author responses from that submission.

Round 1

Reviewer 1 Report

This manuscript proposed that using multiphoton, SHG, and FLIM microscopy to perform a label-free method to determine the liver regenerative potential in the presence of steatosis and fibrosis. Several criteria are indicated such as the zones of the NAD(P)H in autofluorescence images and the changes of lifetime weightings. The manuscript shows detail analysis and discussion. The authors explain the possible reasons about the signal changes relating to certain symptoms. However, there are some questions and comments are addressed as follows:

Questions:

1.     In the captions of Figures 3 (a) and 4 (b), only green color is mentioned to be the SHG of collagen. The blue color is the NAD(P)H autofluorescence. The red color is not clearly mentioned. The authors should explain why there are two colors in SHG images.

2.     In the lifetime analysis, authors focus on the percentage changes of a2 and a3. The a1 value is not mentioned. Since a1 is related to free forms of NADH and a2 is related to bound forms of NADH as mentioned in manuscript. Is the a1 has kind of inversely proportional relation to a2? Since a1 obviously has longer lifetime than a2, it should be easier to have higher contrast a1 value. The authors should comment if the a1 has contribution to the liver analysis or not.

3.     In Figure 3 and 4. In addition to the weighting (percentage) changes of the a1, a2, and a3 in lifetime analysis. The lifetime values have changed as well. Especially the a2 value. In the control, the lifetime of a2 is quite blue indicating longer lifetime compared with the 6th week and 12th week group which has shorter a2 lifetime. Authors should comment about the lifetime changes.

Comments:

1.     In line 39, in Introduction, on Page 1, authors mention the 5-year survival rate only 4-61% [1,2]. I have checked the reference [1,2] and didn’t find that survival rate value. I think the 4% might be a typo. Please confirm it.

2.     In Section 2.2 on Page 3, authors mentioned the formulas for initial liver weight and absolute percentage of liver weight recovery: “absolute weight.” However, the formula definition of “relative weight” is not shown. The formulas are important definitions in Table 1. The formula of “relative weight” should be clearly defined.

3.     In line 19 and 38 on Page 9, line 3 and 4 on Page 10 in Section 3.5, there are no Figure 5 and Figure 6. Please correct the Figure numbers.

Reviewer 2 Report

This manuscript reported a study on assessing the regenerative potential of the liver using fluorescence lifetime imaging microscopy and second harmonic generation. FLIM analysis of NAD(P)H autofluorescence was correlated to the metabolic state of the hepatocytes. Several criteria were identified for indicating reduced liver regenerative potential in the presence of steatosis and fibrosis. The topic is important and interesting.  However, the analysis is not solid and convincing.

1.     FLIM analysis was based on three exponential fitting.  Fig. 3 and 4 displayed mean fluorescence lifetimes, a2 and a3.  However, it was not clear how the lifetimes of decay component 2 of 3 change.  Fitting quality wasn’t shown either. Without the knowledge of t2 and t3, and fitting uncertainty, it was not meaningful to compare a2 and a3 of various samples and claim changes. Moreover, reduces in a2 and a3 mean an increase in a1 (free NADH). However, the manuscript didn’t comment on the changes of free NADH.

2.     3.1 claimed a gradual decrease in the absolute and relative liver weight on the 3rd and 7th days after PH up to the 12th week for steatosis.  This was not true in comparison to normal regeneration. Even for steatosis, apart from week 3, weight on the 3rd day of the weeks 6, 9 and 12 were comparable; weight on the 7th day of the week 9th was lower than that on week 3, 6, 12.

3.     For induced fibrosis, authors claimed that starting from the 4th week, the values of the liver weight even exceeded the corresponding values of physiological regeneration.  However, 3rd day of week 4 (73.1) is smaller than that of normal regeneration (80.9). Authors also claimed that on the 7th day for all stages of the pathology, the percentage of liver recovery was already falling.  But the weight on the 7th day of week 4 was higher than that on the 3rd day.   

Therefore, the manuscript is not suitable for a publication in this journal.